# In Search of Periodicity in the Annual Precipitation in Europe (1881–2020)

Adam Walanus [1], Robert Twardosz [2,*][ID], Marta Cebulska [3] and Arkadiusz Płachta [2]

[1] Geophysics and Environmental Protection, Faculty of Geology, AGH University of Science and Technology, al. Mickiewicza 30, 30-059 Kraków, Poland; walanus@agh.edu.pl

[2] Faculty of Geography and Geology, Jagiellonian University, ul. Gronostajowa 7, 30-387 Kraków, Poland; arekplachta6@gmail.com

[3] Faculty of Environmental and Power Engineering, Cracow University of Technology, ul. Warszawska 24, 31-155 Kraków, Poland; marta.cebulska@pk.edu.pl

[*] Correspondence: r.twardosz@uj.edu.pl

**Abstract:** A new method of searching for periodicity has been developed on the basis of extensive spatio-temporal data. The result, however, produces little more than doubts. The standard Fourier analysis indicates some periods, namely 3.7-, 7.0-, 8.8-, and 17.5-year periods, and these periodic signals are distributed relatively consistently over some regions of Europe. However, the expectations that the exact harmonic 8.8 years of 17.5 years, and not so exact 3.7 years of 7.0 years will be present at the same or close stations are not fulfilled.

**Keywords:** annual precipitation totals; periodic variability; Fourier transform; Europe





## 1. Introduction

As the climate grows warmer [1,2], huge numbers of publications on air temperature changes are being released. The greatest warming of the climate has been observed in Europe [3], and the scenarios of change imply that the warming is bound to intensify [4]. Recent research has shown that average temperatures have been rising to varying degrees from one part of Europe to another [5]. Overall, the warming intensifies towards the northeast of the continent [6]. An increase in the variability of precipitation has been observed in recent decades, which have been exceptionally warm. For example, 2010 was a year of heavy precipitation and disastrous floods in central Europe [7], and 2018 saw a long period of hot and dry weather in many areas of the continent [8–10]. Obviously, there are a number of determinants behind precipitation. In addition to temperature and content of water vapour in the atmosphere, geographical factors also play a role. This multitude of factors of precipitation complicates attempts to establish a link between changes in precipitation and warming [11–13]. This is also mirrored by projections of precipitation amounts modelled on the basis of various scenarios [4].

Long-term precipitation trends are characterised by fluctuations, that is, alternating periods with excess and shortfall of precipitation [14,15]. Naturally, this raises the question whether these fluctuations can have the nature of cyclical changes. In other words, can they be described by statistical methods (by a statistical model)? The search for an answer to such a question is an important issue from the point of view of the forward-looking management of water resources in a given area. Recent decades have seen a noticeable acceleration of the hydrological processes involved in the water cycle in nature as a result of a growing frequency of extreme weather phenomena. To solve water-related problems successfully, the cyclical behaviour of the components of the water cycle needs to be understood [16]. Of course, the issue of cyclicity addressed in this paper is not new, as it dates back to the 19th century [17]. The present research into the cyclicity of precipitation across Europe was inspired by a statistically significant 35-year Brückner cycle of annual precipitation,

produced as a result of research conducted in the Polish Carpathians and their northern foreland [18]. This raises another question, namely one as to whether such long-term annual precipitation cycles also occur in other areas of Europe, or if the latter only see short-term cycles, such as those identified in the literature [18–24], and whether they are statistically significant. Previous Europe-wide research into the cyclicity of precipitation was carried out by [25] on the basis of grid data from the 90-year period 1911–1990. As precipitation is strongly influenced by local conditions, our research relies on measurement data, i.e., data coming directly from weather stations. We focus only on annual totals as the primary precipitation characteristic. Studies of the cyclicity of precipitation have been conducted for large parts of Europe [20,22] and smaller areas [18,23,24,26–31], as well as for individual sites [19,21,32]. Therefore, the findings made by the authors mentioned above pertain to different parts of the European continent and are based on different inputs, usually not comparable in terms of the study timespan. By contrast, this study investigates annual precipitation across Europe for a common 140-year period.

Periodic behaviour of a time series, or its cyclicity, would be an important structure of data, which, in principle, needs to be explained. In the case of our planet, only two periods are obvious: 1 day and 1 year. The moon's rotation does not influence precipitation much, nor does the 11-year solar inner cycle [22,33,34], which, in opposition to two astronomical cycles, is not so precise [35]. However, only periods longer than 2 years are sensible targets to search for in time series of yearly resolution. Periods shorter than 2 years manifest as longer ones by the mechanism known as aliasing.

As many as 60 stations, relatively homogeneously distributed over the full extent of Europe (Figure 1), represented by a 140 year-long time series, provide a significant block of data. Figures 2 and 3 give some idea of its size (Obviously, we are far from what is called "big data"—a thing not available when searching in past records). Figures 2 and 3 show our data simply transformed from the time domain to the frequency domain (see x axis), by the standard method of Fourier transform, which is not a statistical method as there is no loss of information in the transformation. A statistical approach, in which a search is made for significant periods, is applied later. Furthermore, taking into account the geographic relationship between stations is the last step in the analysis. The main result here is a map (Figure 5). While it is impossible to calculate a *p*-value, the map seems to indicate that a periodic structure (T = 7 years) actually exists in the precipitation data. There are also some other periods mentioned below. The method of searching for significant periods is not straightforward because all the available information is made use of. In the end, the results (Figure 7) "suggested" the possibility of an idea, very interesting from the point of view of time series analysis, namely that of harmonic periods. However, a harmonic frequency and the main frequency turned out not to co-exist in the same stations.

## 2. Data and Methods

Precipitation varies greatly in time and space. Therefore, while searching for regularities in long-term precipitation trends, it is necessary to use the longest possible series and as many weather stations as possible. Although precipitation started to be measured at roughly the same time as temperatures in Europe, in most cases, such measurements have not been made continuously and the low density of sites does not allow for precipitation data to be reconstructed and for homogeneous series to be produced. We have had approximately 10 series of monthly precipitation from Europe [36] that run from the 18th century. However, not all of them are continuous. From the beginning of the 19th century, the number of sites which measured precipitation has increased, but systematic measurements at individual stations have only been carried out since the end of the 19th century. In recent years, new secular monthly precipitation series, reconstructed and homogenised from different areas of Europe, have appeared, e.g., [37]. Most of these are available in online databases: *KNMI Climate Explorer* (www.climexp.knmi.nl), *HISTALP* (www.zamg.ac.at), *RIHMI-WDC* (www.meteo.ru), and *ECA&D* (www.ecad.eu) (accessed on 29 January 2020). A range of sources were used to compile a robust database of annual

precipitation totals at the 60 European sites under study for the 1881–2002 research period. These sources included the aforementioned databases, as well as direct inquiries with national meteorological services, principally from Central and Eastern Europe, and with researchers reconstructing and homogenising precipitation data (Figure 1). They are sites with complete and reliable data, used widely in hydroclimatological research. Most of them are lowland stations, i.e., located up to 300 m a.s.l.

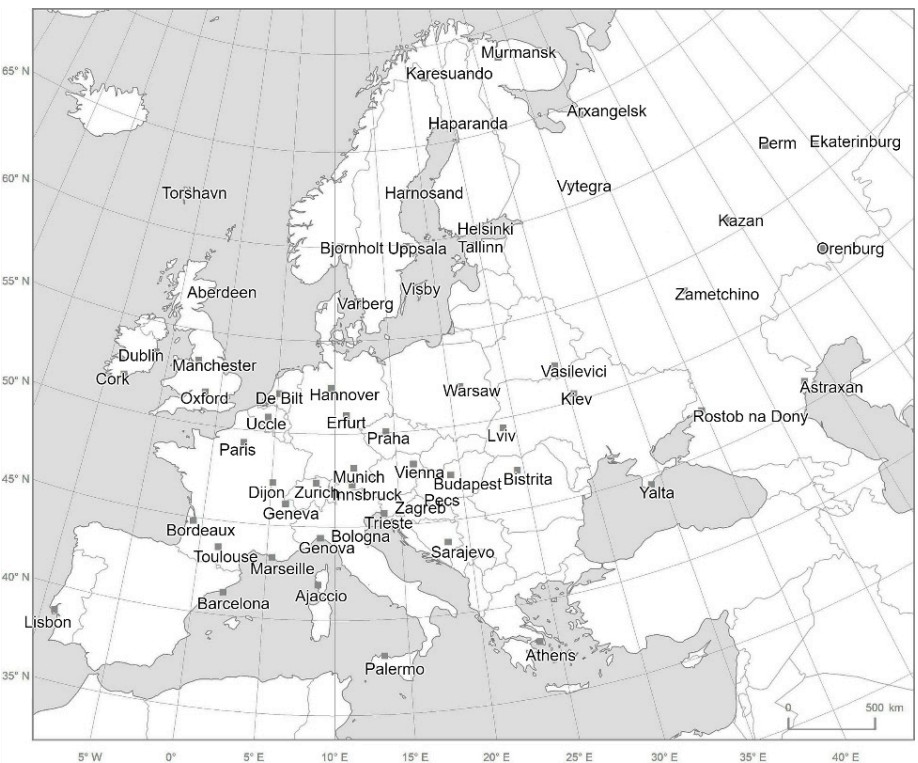

**Figure 1.** Locations of the meteorological stations used in the study.

The method used for searching for periods in the time series is Fourier analysis [38]. The R package was used as a practical tool for performing Fourier transform. Two main lines of R code are presented below which perform Fast Fourier Transform (function: fft) on precipitation data stored in table "d" and save the result in variables "x". The complete code is not much longer.

```
for (i in 1:60){f = fft(d[[i + 1]]); for (j in 1:70){k = (i − 1)*70 + j; xim[k] = Im(f[j]);
                                    xre[k] = Re(f[j])}}
```

Fourier transform is not a statistical method, as there is no loss of information involved; the 140-year long time series (of real numbers, not complex ones) is transformed into two series, each 70 numbers long. One is of amplitudes (A) of periodic "signals", and one is of its phases (p). Periods exhibiting high amplitude are of interest. Phase may be important if there is an independent time series of known phase, for example, solar Wolf numbers. Obviously, the best way of exploiting information about phase is to compare phases between two or more similar time series, which is the case here. If the concrete periodic signal really exists in a geographic area (not "is statistically significant", but "really exists"), it has to exhibit "reasonable" amplitude. Additionally, moreover, the phase for that period should not greatly differ between the sites from that area. Higher and lower precipitation should be found in parallel in neighbouring sites (in the sense of the given period).

So, the data analysis may follow three steps: 1. Searching for periods, in all 60 series, with relatively (not necessarily individually statistically significant) high amplitudes.

2. Checking if the phases for the selected periods are grouping around some value, at least for the same group of sites. 3. Checking, on the map, if the sites indicated by the previous step are geographically close.

In some sense, the notion of "statistical significance" is postponed in the text above. Additionally, or even in opposition to this, the idea of "relative significance" is proposed. Analysing 60 stations with a clear geographical position, but less clear mutual statistical dependence in the sense of precipitation, makes precise *p*-value calculation almost impossible. Furthermore, the key principle of scientific query—objectivity—is not ignored here. After proposing some "relatively" significant periods, the phase (independent parameter on amplitude) is checked, along with its geographical consistency.

## 3. Results and Discussion

Full information resulting from Fourier analysis of 60 precipitation series is presented in two figures. As the series are 140 years long, there are as many as $60 \times 70 = 4200$ points in each. The individual point represents the amplitude (Figure 2) or phase (Figure 3) of the given station (for example Warsaw), for a given period (for example T = 7 years). The horizontal axis of the plot, however, does not represent period but frequency (f = 1/T). The profound nature of Fourier transform is that it resolves frequencies in a linear manner; 1, 2, 3, . . . , 67, 68, 69 cycles per 140 years. This translates to periods hyperbolically; 140, 70, 46.7, . . . , 2.01, 2.06, 2.03 years. To differentiate the values of 2.06 years and 2.03 years is of doubtful sense in nature; however, it is mathematically possible having a series that is 140 years long.

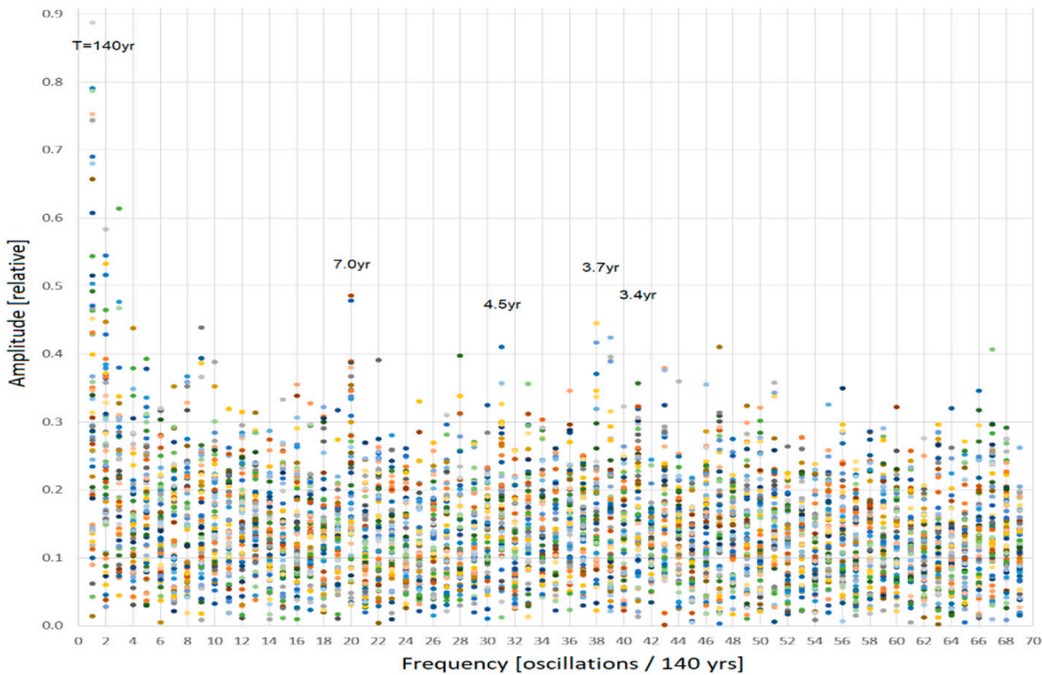

**Figure 2.** Amplitude (in relative units) of periodic signals for all 60 sites. However, colours here indicate the site and are not for practical identification. The horizontal axis represents frequency; some periods are given above the data points.

The vertical axis of Figure 2 needs no comment; it is simply amplitude, in relative units. However, the y axis of Figure 3, where phases are plotted, is worthy of comment. The most natural unit for phase shift is the radian (the natural unit of angle). The range of radians is from 0 to $2\pi$, or from $-\pi$ to $\pi$, as applied in Figure 3. With this figure, one can search for groups of close phases for a given frequency. One needs to be reminded that the phase shift $-\pi$ is equivalent to phase $\pi$. So, the group of points of phase around $\pi$ may manifest themselves as divided, part at the bottom and part at the ceiling of the plot.

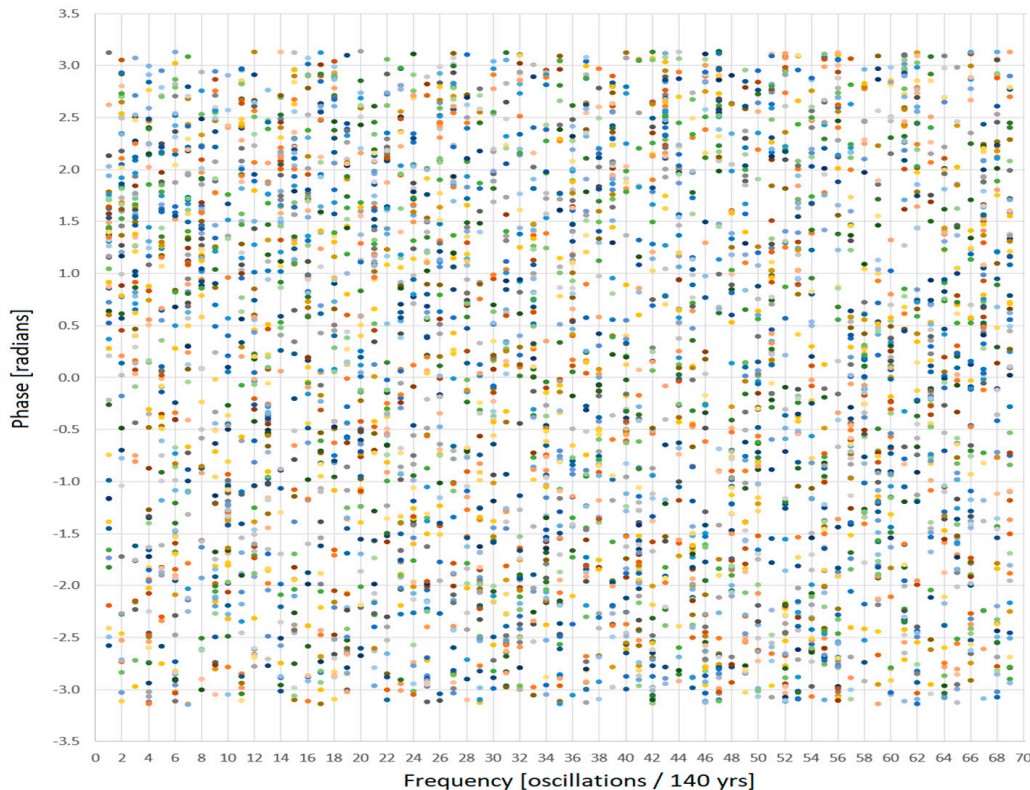

**Figure 3.** Phase of periodic signals for all 60 sites (refer to caption of Figure 2). This phase is cyclic in nature; the value −3.14 is equivalent to +3.14, and the distance between −3.0 and 2.7 is only 0.58 radian, not 5.7 radians.

With two figures, Figures 2 and 3, the following method for searching for significant cycles may be proposed. Search Figure 2 for the frequency (period) for which many stations exhibit high amplitudes. Omitting the case T = 140 years, which is interpreted simply as a trend, the good example is T = 7 years (T = 7 years does not mean a 7-year long low and the next 7 years as a high (Genesis 41:26–36), but rather a 3.5-year low and 3.5-year high). The period of 7 years is equivalent to frequency 20 (7 years = 140 years/20).

Then, in the next step, we examine Figure 3 to see if a group of points is recognisable as an interesting frequency (20). In the case of f = 20, there is group around $p = -0.5$ rad. It is essential that the points forming that group should be those of high amplitude. That question requires the next figure, Figure 4, which is a scatterplot of amplitude plotted against phase for all 60 stations and a chosen frequency (period). In the sample plot obtained for f = 20, two points of similar, maximal amplitude are evident which are, at the same time, of very close phase: pH = −0.8. The question arises: how close are those points geographically? That is easy to answer; however, as there are more interesting points on the plot, a map is necessary.

An attempt to formalise the question of statistical significance of the conclusion considering the presence of a cycle T = 7 years in Europe is to be briefly addressed. In Figure 4, there are six points which are at the same time of the highest amplitude and very close phase. What is the $p$-value of such a situation? The phase span of those six points is less than 0.5, which is 8% of the total phase span ($2\pi$). The first point, that of exactly the highest amplitude, defines the actual value of the phase. What is the probability that the next five special (high amplitude) points fall into the defined span of the phase? As $8\%^5 = 0.0000033$, which is simply the $p$-value, the situation is profoundly significant (not random). Multiplication of the probability of 8% needs the assumption of the statistical independence of the points, i.e., the independence of the data series from different stations, which is obviously

not the case. However, the *p*-value is very small, far below the standard 0.05. Additionally, another (independent) point of view is possible here, that of geographic position.

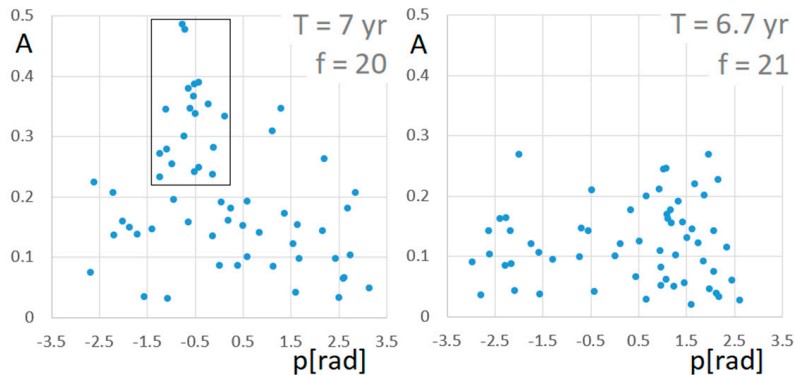

**Figure 4.** Data from Figures 2 and 3 for two given frequencies (periods); the left is for a frequency that seemed to be "significant" and the next a frequency which simply seems to represent the noise alone—the random distribution of phase. For f = 20, the group of stations with high amplitudes and within the narrow span of phases is indicated by the box. It is to be checked to see if the group is also a group on the map.

The isoline map of phases (Figure 5) needs special colouring due to the cyclicity of phases; π is to be represented with a colour similar to the one from the left edge of the scale: −π. However, the case of T = 7 years presents no problem here because phases are concentrated around −0.5, far from −π and π (Figure 4). The isoline map [39] of phases is constructed in the following manner. For each point (E, N) on the map, the average phase is calculated, taking into account all 60 stations, however with different weights. The cyclicity mentioned above, i.e., the fact that −π is equivalent to π, is to be taken into account in the calculation of the average. The phase of the stations selected is weighted by its amplitude. Small amplitudes are of no importance. In practice, the square of the amplitude (the power of the signal) is applied as the weighting parameter. The last point of the algorithm is the calculation of the average for the given map coordinates for all stations. Here, the weighting parameter is the reciprocal of the square of the distance of the calculated point (E, N) from the given site. To avoid infinite weight (a reciprocal of zero) at the point coinciding with any station, an additional distance d is added (d = 100 km). In this manner, it would be easy to avoid infinity: for the location of any station (and its close surroundings), the exact phase coming from that station would be plotted on the map. However, this is not a good option, as the value of the phase is a random variable; by no means is its value to be treated as representative. With parameter d added to the weight, two close stations (for example 20 km apart, while d = 100 km) will produce an area on the map plotted with the colour representing the average phase. A large d (d = 10,000 km) would produce a map of almost uniform colour. A map with a small d (d = 1 km) will show the individual phases of each station. The d actually applied is generally identifiable on the map.

Two areas are generally recognisable on the map presented (Figure 5): western and central Europe and eastern Europe. There are two sites to the west (Lisbon and Barcelona) not following the general trend, as well as Yalta, which is close to the border between the two phases. The three stations identified, and maybe some additional ones (e.g., Astrahan), are to be treated as "necessary" noise in the data set. Ignoring "noise," two values of phase dominate the map: −0.5 to the west and 1.0 to the east. Generally, the map seems to confirm the real (statistically significant) existence of the period T = 7 years. The distribution of colours is relatively regular on the map, producing a sensible pattern. It may be that the pattern is not easy to explain; however, it is far from a purely random one like the distribution of points at the right of Figure 4 (T = 6.7 years).

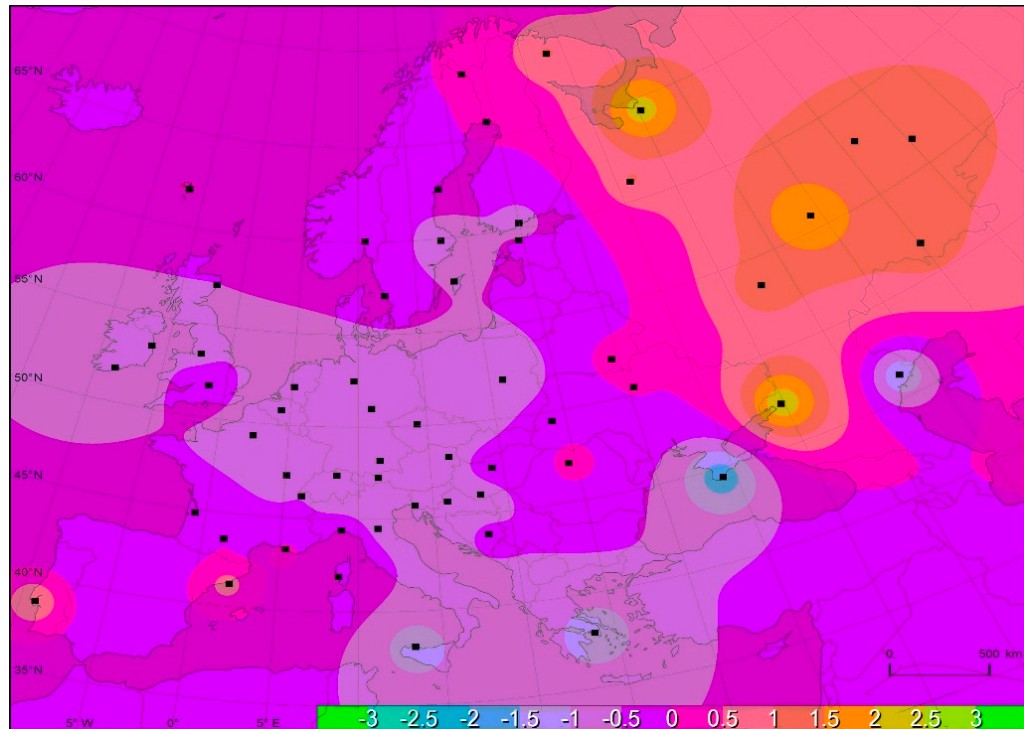

**Figure 5.** Isolines map of a phase of the periodic "signal" of T = 7 years. Weighting by amplitude is applied. The smoothing parameter is d = 100 km. The colouring is cyclic, which is not essential here as the prevailing phase is far from $-\pi$ and $\pi$.

In Figure 2 there are other periods mentioned, not simply T = 7 years. Reading that figure is not easy, especially given that every case of a frequency suspected of being represented by many high amplitudes is to be compared to see if phases connected with high amplitudes are consistent, which would need a figure of type Figure 4. To analyse all 69 frequencies by means of separate scatter plots (Figure 4) would be troublesome and full of subjectivity. The scatterplots are to be converted to one number each. The final plot is Figures 6 and 7. The vertical axis on Figure 7 represents the number of points n (number of stations). The method is as follows: the 60 values of amplitudes, for a given frequency f, are sorted in descending order. Then, the standard deviation of phase of these points (stations) is calculated for two, three, four, and so on, uppermost points. An example of the resulting series of SDs is presented in Figure 6. This is not a typical example; it is the case for T = 7 years, which is already accepted as significant. A really random case of T = 6.7 years produces a horizontal line. If the points of high amplitudes are, at the same time, of similar phase, the standard deviation of phase should be low for those n points. Which value of SD is low? The standard deviation of a random variable uniformly distributed between $-\pi$ and $\pi$ is equal to $2\pi/(12^{1/2}) = 1.81$. Let us accept as "low" half of that value, i.e., 0.91. The natural number plotted in Figure 7 is the maximal number of points of the highest amplitudes, for which the standard deviation of phases does not exceed 0.91. Both plots in Figure 4 are transformed to the single numbers n = 17 and n = 1, respectively, so the method seems to work well, as n = 17 is evidently higher than n = 1, and so the special pattern of points in the left part of Figure 4 is well reproduced in one number.

The plot in Figure 6 is not monotonic, so the automated use of the criterion <0.91, going from left to right, may not produce the best estimate of the number of sites. In analysing only four interesting frequencies (not all 69) we "smoothed" the plot in searching for the "real" number of sites, for which there is a standard deviation less than 0.91. The sites extracted that way are used in Figure 8.

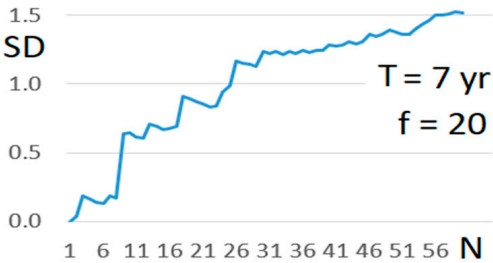

**Figure 6.** Consecutive standard deviations of phases for sites of amplitudes in descending order (see text). Here, as many as n = 17 sites of the highest amplitude produce SD < 0.91. The value of n is used in Figure 7.

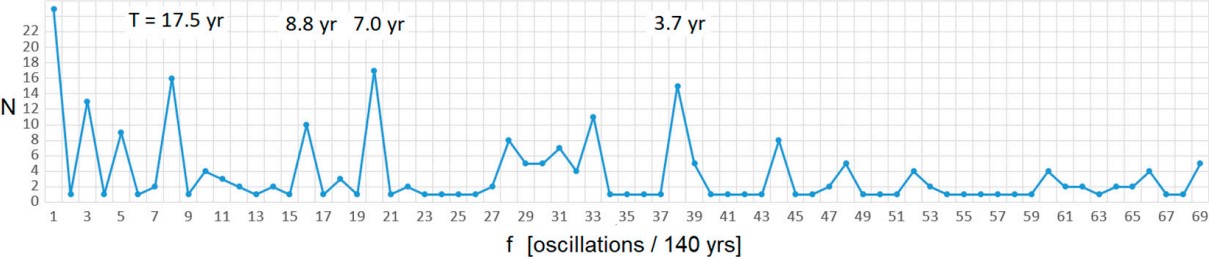

**Figure 7.** Number n of stations with highest amplitude and close phases at the same time (see Figure 6 and description in the text). Ignoring f = 1 (trend, and also the low f = 3), three n values are evidently the highest. The fourth (f = 16) is accepted because it is a harmonic of f = 8 (f = 33 is not such a case).

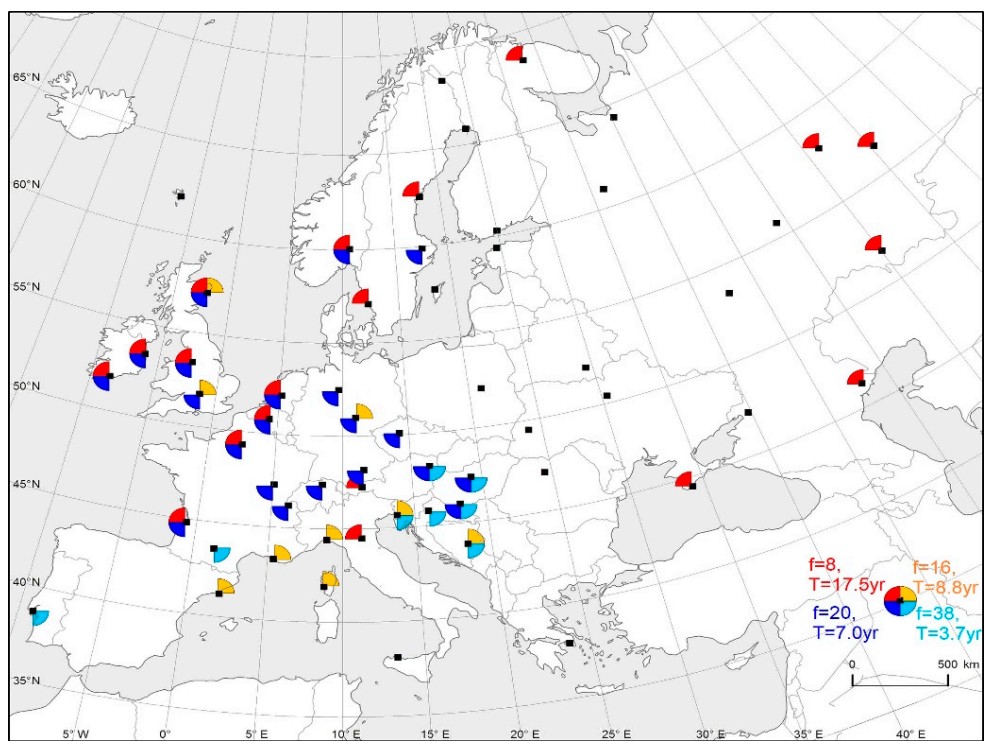

**Figure 8.** With four colours for different quarters, four frequencies are marked if present at the station: red and yellow for f = 8 and f = 16, respectively, and dark and light blue for f = 20 and f = 38, respectively (periods are, respectively: T = 17.5 years, 8.8 years, 7.0 years, and 3.7 years). Two pairs of harmonically connected frequencies (f = 8 and 16; and f = 20 and 38 ≈ 40) are indicated if present at a given site. The result is negative; harmonics are generally at other sites than its respective main frequency. However, some spatial uniformity of frequency distribution is visible.

The resulting Figure 7 is self-explanatory. Ignoring low frequencies (1, 3, or 5 per 140 years), four other frequencies can easily be identified (they are described by the periods on the figure). It is astonishing that there are two pairs of frequencies and their harmonics. Frequency f = 16 is exactly the first harmonic of f = 8 (T = 17.5 years), frequency f = 38 is not so precisely harmonised with f = 20.

In fact, there is another, independent point of view to check the proposition that there are two pairs of frequencies, namely that of the geographic position of stations.

The map obtained (Figure 8), however, contradicts the idea of harmonic frequencies. More precisely, the harmonics f = 8 and f = 16, marked by red and yellow quarters, respectively, are almost completely divergent. The quasi-harmonics f = 20 and f = 38, marked by blue (dark and light, respectively), are slightly better, as as many as three sites in the southern part of central Europe and the Balkan stations exhibit both frequencies. Neither colour, however, seems to be randomly distributed, which suggests that the four frequencies presented have statistical significance. Those three stations provide a good example of non-randomness (low *p*-value). Having three among the 60 cases possessing some feature, the following probabilistic reasoning provides the *p*-value: the first station defines the geographic position, the probability is that the second one is closest to the defined position being 1/59. Similarly with the third case; the final *p*-value = (1/59) × (1/58) = 0.0003, which is much lower than the standard significance level α = 1/20 = 0.05.

The idea of harmonic frequency dominates music. Violin strings produce many harmonic sounds, and neighbouring strings are harmonically tuned to each other; there is no other way. The sine function is fundamental in physics (and music). Maybe nature (precipitation) is more complicated. We have absolutely no idea what kind of oscillating "string" may oscillate with a period 17.5 years. Here, we refer only to the result of the Fourier analysis of data. The analysis presented is of a relatively good level of objectivity. All steps are presented. It is not easy to completely, objectively prove that "the final map" (Figure 8) is random or not. However, some features are obvious (one is already mentioned above). For example, half of the area on the map, from 20° E to the east, contains one only one exclusive frequency: "red", f = 8, T = 17.5 years. A similar division is possible for "light blue", f = 38, T = 3.7 years. Virtually all sites exhibiting that frequency are south of 50° N (or even 48° N) and west of 20° E. Additionally, the sites with no periodicity at all are spatially correlated, creating a striking hole on the map with no colour. It is formed by a vast area stretching from the eastern part of central Europe to the western part of European Russia.

The divergence of pairs of harmonic oscillations that has been noted may be addressed statistically. There are 60 stations, 19 of which are "red" and 9 are "yellow", while just one is both "red" and "yellow". The *p*-value is to be calculated by applying the binomial distribution with parameters: number of trials N = 9, and probability of success *p* = 19/60 = 0.32. The probability of exactly one "success" is 0.14. In the case of the null hypothesis that there is some mechanism separating the "red" and "yellow" cases, the case of zero coincidences should also be added (0.14 + 0.032). However, already the specific case of exactly one coincidence is very probable (0.14 > 0.05), so the result is that both frequencies, f = 8 and f = 16 are distributed independently, and exhibit no spatial correlation. It is self-evidently a negative result. The case of "blue" frequencies (f = 20 and f = 38) has already been treated. The evident spatial correlation of coincident harmonics (in the southern part of central Europe and the Balkans) can be explained simply by the correlation of close stations.

The quasi-harmonics f = 20 and f = 38 are only "quasi". The frequency f = 20 is absolutely maximal (excluding f = 1) on Figure 7. Its neighbouring frequencies are rated at minimal value or close to minimal. Its exact first harmonic f = 40 is zero, f = 39 is almost zero, so the idea of "quasi" harmonised f = 38 can simply be rejected. A worse (better?) situation is the situation with f = 8 and f = 2 × 8 = 16, however, the amplitude of the peak at f = 16 (Figure 7) is not so sharp. The f = 33, not considered here at all, is one step higher, as is the case for f = 3. (The "numerical coincidence" of 33 and 3 has no significance, however this is not the case with 8 and 16.). The lack of coincidence of occurrence of f = 8 and f = 16 at the stations leaves no doubt. There is no observed periodicity of T = 17.5 years, of non-

harmonic shape being the sum of two sine time series. The idea of harmonic composition is to be definitely rejected (it is a negative result, however it is a result). What is left? The non-random spatial distribution of the four (independent) periodicities considered.

Initially, it can be seen that there are areas in Europe with two significant annual precipitation cycles. In the westernmost parts of Europe, the cycles are as follows: T = 17.5 years and T = 7.0 years—red and dark blue, respectively. In the southern part of central Europe and in the Balkans, they are the following cycles: T = 7.0 or 8.8 years (dark blue or yellow, respectively) and T = 3.7 years (light blue). Similarly, in Croatia, Gajić-Čapka (1994) found quasi-periodic oscillations of 2.2 years and 4.7 years using spectrum power analysis. At less than half of the stations, only one statistically significant cycle occurs. Therefore, T = 8.8 years is the only cycle found at sites in the north-western Mediterranean (yellow), T = 7.0 years in the Alpine countries (dark blue), and T = 17.5 years in the south-eastern part of Europe and at some sites in Scandinavia (red). Using spectral analysis, Maheras et al. (1992) identified short-term precipitation cycles of 13.6, 3.5, and 2.2 years in the Mediterranean region. Qian et al. (2000) found quasi-periodic oscillations with periods of 23.8 months (2.0 years) and 43.5 months (3.6 years), respectively, with the 3.6-year oscillation likely to have been related to the atmospheric circulation from the North Atlantic and Europe. In summary, short-term cycles prevail in Europe. According to research by Ilyés et al. (2017, 2018), there are sites which record as many as a dozen or so such short-term cycles (16–17 at Hungarian sites). According to these authors, the longest rainfall cycles evolve, i.e., they become shorter and shorter, which, as they believe, may be related to the increasing variability of the weather.

## 4. Conclusions

The present study has sought to identify cyclical changes in annual precipitation from 60 meteorological stations in Europe over the 140-year period 1881–2020. Based on the method of Fourier analysis, research has found short-term cycles of 3.7, 7.0, 8.8, and 17.5 years. It is difficult to identify the underlying causes. Cyclical fluctuations with a period of less than 10 years prevail in the research area, especially in southern Europe, and those with a period of 17.5 years at some sites in northern Europe and at the periphery of south-eastern Europe. The study has found that there are no significant annual precipitation cycles in around half of Europe, mainly east of 20° E.

The shortest precipitation cycle identified (3.7 years) may be the result of the North Atlantic Oscillation, which was mentioned above by reference to the relevant literature. It is difficult to propose any causal mechanism for the remaining cycles.

The method of verification of significance of results presented is rather atypical, however, and not very sophisticated in a mathematical sense. Its essence lies in a massive Fourier analysis of many time series, not only in the aspect of amplitude or power (which is typical), but also of the phase of the sine wave. Having as many as 60 time series (which are not very short) is a powerful tool of statistical analysis. That is the reason why the authors treated the strict question of the *p*-value calculation relatively lightly. The more formal reason for that lies in the difficulties associated with the mutual, geographical dependence of precipitation series. If the method of checking statistical significance proposed in Figure 6 may not have been deeply convincing, its result (Figure 7) is not affected by this concern. The vertical axis of Figure 7 consists of natural numbers (number of counts), which makes for an easy strict *p*-value calculation based on the binomial probability distribution.

Finding the couple of significant frequencies strictly connected by the harmonic relation (8 and 16), as well as an approximate harmonic (20, 38) in Figure 7, raised high expectations of the possibility of perfect confirmation of the periodicities in the precipitation series. The facts (Figure 8), however, closed down this question, leaving no doubt. The frequency 16 exists in different stations to that of 8 oscillations per 140 years. We decided not to examine the question of harmonics at all, but to present what happened in our investigations.

The most essential result here is the isoline map of the phases of one period T = 7 years (Figure 5), which presents a clear, easily recognised pattern. Contrary to the question of harmonics, we have not presented here the results of checking the statistical significance of the pattern by the bootstrapping method, because it needs too much space to present (rather boring) figures. The geographical pattern visible on the map, seemingly natural, needs explanation. However, above all, of course, the value T = 7 years (and the others from Figure 7) need an explanation in some natural cause.

**Author Contributions:** Conceptualization, A.W.; methodology, A.W.; formal analysis, A.W., R.T. and M.C.; investigation, A.W., M.C. and R.T.; data curation, R.T. and A.P.; writing—original draft preparation, A.W., R.T. and M.C.; writing—review and editing, A.W., R.T. and M.C.; and supervision, A.W. All authors have read and agreed to the published version of the manuscript.

**Funding:** This research received no external funding.

**Institutional Review Board Statement:** Not applicable.

**Informed Consent Statement:** Not applicable.

**Data Availability Statement:** Not applicable.

**Acknowledgments:** We thank Paweł Pilch and Martin Cahn for reviewing the English. We thank Davide Scafidi for making precipitation data from the Historical Meteorological Observatory of the University of Genoa available.

**Conflicts of Interest:** The authors declare no conflict of interest.

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
