# Peer review of "In Search of Periodicity in the Annual Precipitation in Europe (1881–2020)"

_water, doi:10.3390/w14132026_

Round 1

Reviewer 1 Report

Literature review is weak and needs to be upgraded using other researches. Also, the conclusion should be revised based on comparing results with other researches. 

Statistical analysis of the quality of data or missing data should be discussed. 

The methodology is not clear.

“The period of 7 years is equivalent to frequency 20 (7 yr=140 yr/20)” means 20 * 7-years periods? Thus, the is a difference between this value (20) and the definition of frequency of occurring or the definition of probability. why 20 has been selected?

Please use better writing, some texts are not such as a scientific manuscript:  “The vertical axis of Fig.2 needs no comment, it is simply amplitude, in relative units. However, the y axis of Fig.3, where phases are plotted, is worthy of comment”. Line 326: what is NAO, Line 327: “mentioned above by reference“ is vague

Figures 6 and 7 should be redrawn. All figures should have perfect axis names and legend.

How it is possible to search for trends of drought or above-normal conditions, certainly, these trends deviate normal conditions for estimating a constant or periodicity periods.

Author Response

We would like to sincerely thank the Reviewers for their insightful reviews of our article. We have corrected the article as suggested.

Please find below our detailed answers

***************************************************************************

Suggestions for Authors

  • Literature review is weak and needs to be upgraded using other researches. Also, the conclusion should be revised based on comparing results with other researches.

Authors: Introduction is extended.  6 additional articles were cited

  • Statistical analysis of the quality of data or missing data should be discussed. 

Authors: In this study, we did not deal with the issue of statistical data quality analysis, because the article uses only complete series of atmospheric precipitation, reconstructed and already checked for homogeneity. This is mentioned in the article.

  • The methodology is not clear.

Authors: The methodology is not standard one. It makes use of all data extracted from Fourier analysis, including phase, what is not typical in climate study. The number od stations of definite geographic relation is large, what need some compression of results (Figs. 6, 7), which is not typical as well. Deep methodological explanations would consume all space for climatic interpretations, which are essential, anyway.

Supplements were introduced in the final part of the introduction - also in accordance with the remark of the second reviewer.

  • “The period of 7 years is equivalent to frequency 20 (7 yr=140 yr/20)” means 20 * 7-years periods? Thus, the is a difference between this value (20) and the definition of frequency of occurring or the definition of probability. why 20 has been selected?

Authors: Frequency 20 means 20 oscillations (periods) per (within) time series length (140 yr). The value of 20 is chosen because it produces the maximal (despite 1) peak in Fig. 7.

  • Please use better writing, some texts are not such as a scientific manuscript:  “The vertical axis of Fig.2 needs no comment, it is simply amplitude, in relative units. However, the y axis of Fig.3, where phases are plotted, is worthy of comment”. Line 326: what is NAO, Line 327: “mentioned above by reference“ is vague

Authors: The y axis in Fig. 3 is that of phase, as described. For the given stations and given frequency one can read out the phase of the signal, similarly to amplitude in Fig. 2. Thus, from the data from both figures one can (principally) reproduce the sinus function with parameters, for each station and each frequency.

NAO - North Atlantic Oscillation

  • Figures 6 and 7 should be redrawn. All figures should have perfect axis names and legend.

Authors: Yes, its true, the trade off between saving space and figure clarity is probably moved too much to savings. May be larger “N” and “f” in Fig. 7 would be enough correction.

  • How it is possible to search for trends of drought or above-normal conditions, certainly, these trends deviate normal conditions for estimating a constant or periodicity periods.

Authors: Sorry, do not understand.

Reviewer 2 Report

REVIEWER’S COMMENTS TO AUTHOR:

In this paper, the authors focus on the Fourier Transform analysis of annual precipitations over a large set of stations in Europe. They present a new and interesting method of analysis based on searching for periods of high amplitudes with similar phases and found that periods of 3.7, 7.0, 8.8, and 17.5 yr are the most relevant. These results can contribute to a better understanding of the climate of precipitation in Europe, particularly about the short-term cycles.

General comments:

In general, this manuscript is well argued and contains the main sections required by the journal guidelines. However, it contains some errors and lacks some necessary explanations at various points in the text and several figures should be improved. For this reason, I consider that a revision is necessary before this paper can be considered for publication in this journal. Some minor corrections related to the spelling and grammar are also necessary. So, I suggest that the authors carefully revise the text to correct these issues.

·       The citation to ref. [23] is missing in the text.

·       The introduction section must clearly include the main hypothesis for which the authors propose a new method for searching periodicities in annual precipitation, giving a brief notion that anticipate the novelty of this method.

·       Lines 81-85 are difficult to understand. Particularly, the expression “Based on […] and some of the researchers who have dealt with the reconstruction and homogenisation of precipitation, […]” should be checked.

·       In regard to the previous point, authors state that some data were obtained directly from national meteorological services in line 82. However, there is no allusion to their links or to the states where those meteorological services belong. If the number of these organizations is too high, we suggest that authors mention at least the main areas that they cover.

·       The Figures 1 and 5 should be submitted with more resolution or authors should use bigger font size to make the name of locations, coordinates and scale more readable.

·       In line 93, it is claimed that “Two main lines of R code are presented below which perform “fft” on precipitation data […]”. To the best of my knowledge, fft is the acronym of the “Fast Fourier Transform” algorithm. The full name of the algorithm should be explicitly mentioned, and the acronym should be in parenthesis, as it is the first time that the authors allude to it.

·       The portion of code shown in line 95 should perhaps be separated into several lines to make it more readable in a similar way to other studies, such as the following:

Tsoukalas, I.; Kossieris, P.; Makropoulos, C. Simulation of Non-Gaussian Correlated Random Variables, Stochastic Processes and Random Fields: Introducing the anySim R-Package for Environmental Applications and Beyond. Water 2020, 12, 1645. https://doi.org/10.3390/w12061645

·       Lines 102-106 are difficult to understand. The most confusing lines are 104-106, where the argument of authors is poorly understandable. These sentences should be revised. What do the authors mean exactly? Basing on the next lines I guess that they mean that if an exact periodic signal really exists in an area, the time series from stations in that area should exhibit similar phases.

·       Figures 2 and 3 should be placed consecutively because they were both cited close together. Moreover, they should have a legend including the colour for each station. For the sake of clarity, authors can make one multi-panel figure which contains both figures and the legend just once. Due to the large set of stations depicted (60), if the complete set is too large that it hides a relevant part of the plot, authors should consider including at least the most prominent stations, such as those commented in the text.

·       In the caption of Figure 3 in line 157 it is claimed that the phase is cyclic in nature and the authors give two “examples”. The phase is in the range [-pi,pi) and the first “example” where it is stated that -3.14 and 3.14 are equivalent is correct, but the second statement is false. Any value in this interval is univocal and both -3.0 and 2.7 belong to this interval. Therefore, the expression “-3.0 is close to +2.7” has no sense in the same terms. An example of two equivalent phases would require one value shifted from another in exactly a multiple of 2pi or 6.28 radians, e.g.: -1.57 and 4.71, which are shifted by 6.28 rad (one cycle or period), are equivalent. Other possibility can be searching for half-periods, which might be the possible reasoning considered by the authors. In this case, two values separated by a half-period must be shifted by exactly pi or 3.14 radians.

·       In line 219-220 it is stated that “How both plots in Fig.4 are transformed to n=17 and n=1 is described below”, but Fig. 4 only has one plot, the isolines map. These lines must be revised. In fact, all text in the lines 218-232 is confusing because firstly some results and the plot obtained are cited and then the method used is said to “work well” without explaining the process or method of how they are computed. The reader only knows that “The scatterplots are to be converted to one number each” but not how they are converted into numbers or where these numbers come from until he/she reaches the line 220 and subsequent lines. For this reason, this paragraph must be checked to firstly explain the method of computation and next the outcomes obtained through this method.

·       Regarding the previous point, Figure 7 is cited before Figure 6. Nevertheless, since the order of the method of computation (related to the clarifying example from Figure 6) and results (included in Figure 7) is inverted and must be changed, the final order of citation and numbers of these figures should perhaps remain unmodified. Anyways, both figures should appear right after the paragraph where they are first cited.

·       The explanation of Figure 8 could be improved if authors considered to include the chosen criterion to affirm that one period is “present” or can be ignored at each station. I guess that the box criterion from Figure 4 has been chosen to decide if it can be associated or discarded the 7 yr (f=20) period to each station, but this should be explicitly mentioned. Also, if the same criterion has been used for the rest of periods, it would be appropriate to mention at least the minimum threshold for the amplitude and the span of phases with the maximum interval found in these stations.

·       A larger version of Figure 8 with a bigger font size and more resolution should be provided to make it more readable. A legend should also be included, avoiding the need for explaining the colour of each quarter/period in the caption and every time that the results yielded from the map are commented. This figure should also appear right after the paragraph where it is first cited.

·       I suggest revising the calculations on lines 280-282. The probability of exactly one success for a binomial distribution with the mentioned parameters is a bit lower. If one is more cautious with rounding errors, the probability of zero successes can also vary a bit. However, the argument is still valid.

·       In lines 292-293, authors base on the Fig. 7 to state that some frequencies are zero. Considering that figure, the minimum y-axis value is 1, as they indeed remark at the end of line 292. For this reason, it is not clear what the authors mean when they associate zero values to some frequencies. Is it possible that they are discussing about co-occurrence? In that case, it might be understandable the zero values, but an explicit commentary should be added.

·       After seeing the Figure 5 with the isolines map of phases for the 7 yr periods, one might wonder whether the other three prominent periods (17.5, 8.8 and 3.7 yrs) exhibit similar patterns. Did the authors check the isolines maps of these other relevant periods? Is there any conclusive geographical pattern, similar or different, for those periods? A commentary about this should be included in the discussion of results and, if they lead to different relevant results from the 7 yr period, with their corresponding figures. Also, if it is appropriate and relevant conclusions are obtained, it should be mentioned in the conclusions section.

·       Line 297 is too informal, more specifically, after the opening parenthesis. I suggest rephrasing this expression.

·       Lastly, in the conclusions section in line 346, the authors claim that the most essential result found in their research is the isoline map of the phases for period T = 7yr (Figure 5). However, this result is not mentioned in the abstract. I suggest including it briefly in the abstract.

Author Response

We would like to sincerely thank the Reviewers for their insightful reviews of our article. We have corrected the article as suggested.

Please find below our detailed answers

**************************************************************************

REVIEWER’S COMMENTS TO AUTHOR:

In this paper, the authors focus on the Fourier Transform analysis of annual precipitations over a large set of stations in Europe. They present a new and interesting method of analysis based on searching for periods of high amplitudes with similar phases and found that periods of 3.7, 7.0, 8.8, and 17.5 yr are the most relevant. These results can contribute to a better understanding of the climate of precipitation in Europe, particularly about the short-term cycles.

General comments:

In general, this manuscript is well argued and contains the main sections required by the journal guidelines. However, it contains some errors and lacks some necessary explanations at various points in the text and several figures should be improved. For this reason, I consider that a revision is necessary before this paper can be considered for publication in this journal. Some minor corrections related to the spelling and grammar are also necessary. So, I suggest that the authors carefully revise the text to correct these issues.

Authors: The article has been revised in terms of language by a native speaker

2.1.     The citation to ref. [23] is missing in the text.

Authors: changed

2.2. The introduction section must clearly include the main hypothesis for which the authors propose a new method for searching periodicities in annual precipitation, giving a brief notion that anticipate the novelty of this method.

Authors: Introduction is extended.

2.3.  Lines 81-85 are difficult to understand. Particularly, the expression “Based on […] and some of the researchers who have dealt with the reconstruction and homogenisation of precipitation, […]” should be checked.

Authors: changed

2.4.   In regard to the previous point, authors state that some data were obtained directly from national meteorological services in line 82. However, there is no allusion to their links or to the states where those meteorological services belong. If the number of these organizations is too high, we suggest that authors mention at least the main areas that they cover.

Authors: changed 

2.5.    The Figures 1 and 5 should be submitted with more resolution or authors should use bigger font size to make the name of locations, coordinates and scale more readable.

Authors: Fig. 1- is changed, Fig .5 – really the numbers of legend are too small? Coordinates are really small if present at all, however, it is simply Europe.

2.6.  In line 93, it is claimed that “Two main lines of R code are presented below which perform “fft” on precipitation data […]”. To the best of my knowledge, fft is the acronym of the “Fast Fourier Transform” algorithm. The full name of the algorithm should be explicitly mentioned, and the acronym should be in parenthesis, as it is the first time that the authors allude to it.

Authors: changed

2.7.  The portion of code shown in line 95 should perhaps be separated into several lines to make it more readable in a similar way to other studies, such as the following:

Tsoukalas, I.; Kossieris, P.; Makropoulos, C. Simulation of Non-Gaussian Correlated Random Variables, Stochastic Processes and Random Fields: Introducing the anySim R-Package for Environmental Applications and Beyond. Water 2020, 12, 1645. https://doi.org/10.3390/w12061645\

Authors: line 95: it is much better, if possible, to have all body of “for” loop in one line. The piece of code is included only to give to the reader some “visual” idea of how it may look like. The reader may contact authors for further explanations.

2.8. Lines 102-106 are difficult to understand. The most confusing lines are 104-106, where the argument of authors is poorly understandable. These sentences should be revised. What do the authors mean exactly? Basing on the next lines I guess that they mean that if an exact periodic signal really exists in an area, the time series from stations in that area should exhibit similar phases.

Authors: lines 104-106 are changed

2.9.       Figures 2 and 3 should be placed consecutively because they were both cited close together. Moreover, they should have a legend including the colour for each station. For the sake of clarity, authors can make one multi-panel figure which contains both figures and the legend just once. Due to the large set of stations depicted (60), if the complete set is too large that it hides a relevant part of the plot, authors should consider including at least the most prominent stations, such as those commented in the text.

Authors: The points are in different colors only to present the idea where the sites are on the plot. It is impossible (as it is stated in figure caption) to identify sites by color because colors are too similar, because it is so many sites. To have both figures closer is good idea, if editor have nothing against.

2.10.      In the caption of Figure 3 in line 157 it is claimed that the phase is cyclic in nature and the authors give two “examples”. The phase is in the range [-pi,pi) and the first “example” where it is stated that -3.14 and 3.14 are equivalent is correct, but the second statement is false. Any value in this interval is univocal and both -3.0 and 2.7 belong to this interval. Therefore, the expression “-3.0 is close to +2.7” has no sense in the same terms. An example of two equivalent phases would require one value shifted from another in exactly a multiple of 2pi or 6.28 radians, e.g.: -1.57 and 4.71, which are shifted by 6.28 rad (one cycle or period), are equivalent. Other possibility can be searching for half-periods, which might be the possible reasoning considered by the authors. In this case, two values separated by a half-period must be shifted by exactly pi or 3.14 radians.

Authors: “-3.0 is close to +2.7”  is changed

2.11.  In line 219-220 it is stated that “How both plots in Fig.4 are transformed to n=17 and n=1 is described below”, but Fig. 4 only has one plot, the isolines map. These lines must be revised. In fact, all text in the lines 218-232 is confusing because firstly some results and the plot obtained are cited and then the method used is said to “work well” without explaining the process or method of how they are computed. The reader only knows that “The scatterplots are to be converted to one number each” but not how they are converted into numbers or where these numbers come from until he/she reaches the line 220 and subsequent lines. For this reason, this paragraph must be checked to firstly explain the method of computation and next the outcomes obtained through this method.

Authors: “but Fig. 4 only has one plot, the isolines map.” – actually no, it is Fig. 5.

Text is reorganized.

2.12.   Regarding the previous point, Figure 7 is cited before Figure 6. Nevertheless, since the order of the method of computation (related to the clarifying example from Figure 6) and results (included in Figure 7) is inverted and must be changed, the final order of citation and numbers of these figures should perhaps remain unmodified. Anyways, both figures should appear right after the paragraph where they are first cited.

Authors: changed

2.13.       The explanation of Figure 8 could be improved if authors considered to include the chosen criterion to affirm that one period is “present” or can be ignored at each station. I guess that the box criterion from Figure 4 has been chosen to decide if it can be associated or discarded the 7 yr (f=20) period to each station, but this should be explicitly mentioned. Also, if the same criterion has been used for the rest of periods, it would be appropriate to mention at least the minimum threshold for the amplitude and the span of phases with the maximum interval found in these stations.

Authors: Explanation is added.

2.14.       A larger version of Figure 8 with a bigger font size and more resolution should be provided to make it more readable. A legend should also be included, avoiding the need for explaining the colour of each quarter/period in the caption and every time that the results yielded from the map are commented. This figure should also appear right after the paragraph where it is first cited.

Authors: Legend is added to the figure, caption is changed.

2.15.       I suggest revising the calculations on lines 280-282. The probability of exactly one success for a binomial distribution with the mentioned parameters is a bit lower. If one is more cautious with rounding errors, the probability of zero successes can also vary a bit. However, the argument is still valid.

Authors: Exactly true. Cumulative probability has been calculated. 0.16 <- 0.14; 0.031 <- 0.032.

2.16.       In lines 292-293, authors base on the Fig. 7 to state that some frequencies are zero. Considering that figure, the minimum y-axis value is 1, as they indeed remark at the end of line 292. For this reason, it is not clear what the authors mean when they associate zero values to some frequencies. Is it possible that they are discussing about co-occurrence? In that case, it might be understandable the zero values, but an explicit commentary should be added.

Authors: Text is changed.

2.17.       After seeing the Figure 5 with the isolines map of phases for the 7 yr periods, one might wonder whether the other three prominent periods (17.5, 8.8 and 3.7 yrs) exhibit similar patterns. Did the authors check the isolines maps of these other relevant periods? Is there any conclusive geographical pattern, similar or different, for those periods? A commentary about this should be included in the discussion of results and, if they lead to different relevant results from the 7 yr period, with their corresponding figures. Also, if it is appropriate and relevant conclusions are obtained, it should be mentioned in the conclusions section.

Authors: Map in Fig. 8 is not of isoline type, however, gives some insight into the question of geographical consistency of all four periods. Caption is modified (see 2.14).

2.18.       Line 297 is too informal, more specifically, after the opening parenthesis. I suggest rephrasing this expression.

Authors: changed

2.19.      Lastly, in the conclusions section in line 346, the authors claim that the most essential result found in their research is the isoline map of the phases for period T = 7yr (Figure 5). However, this result is not mentioned in the abstract. I suggest including it briefly in the abstract.

Authors: “Abstract: … periodic signals are distributed relatively consistently over some regions of Europe. … “

Abstract is rather short, adding only Fig. 5 reference is not elegant…?

Round 2

Reviewer 1 Report

-

Author Response

Without comments

Reviewer 2 Report

The manuscript has been improved considerably. Nevertheless, some minor corrections are still necessary.

2.11: Fig. 4 truly includes two plots, as the authors answered in response to point 2.11. I confounded that figure with Fig. 5. The paragraph in lines 241-263 has been reorganized and the text is more understandable. However, the old lines (248-250) remain and the text is redundant (they repeat the same idea as in lines 261-263).

2.15: The calculations of the probability of the binomial distribution were corrected and satisfactorily answered in the response to point 2.15. However, changes in the text were only made in the line 321. Lines 323 and 324 (results in parenthesis) have not been changed yet.

Author Response

Here is the answer to the Reviewer's comments. They are in green in the article

2.11: Fig. 4 truly includes two plots, as the authors answered in response to point 2.11. I confounded that figure with Fig. 5. The paragraph in lines 241-263 has been reorganized and the text is more understandable. However, the old lines (248-250) remain and the text is redundant (they repeat the same idea as in lines 261-263).

Authors: Please excuse us for such a silly error. We try to believe that it is up to some printing imp.

2.15: The calculations of the probability of the binomial distribution were corrected and satisfactorily answered in the response to point 2.15. However, changes in the text were only made in the line 321. Lines 323 and 324 (results in parenthesis) have not been changed yet.

Authors: The printing imp gets here its proper face. I remember exactly the single figures to be corrected and, may by not marked at all, and the corresponding author missed it. Excuse us again.